# Molecular determinants of Neu5Ac binding to a tripartite ATP independent periplasmic (TRAP) transporter

**Parveen Goyal[1,2,3]\*, KanagaVijayan Dhanabalan[4], Mariafrancesca Scalise[5], Rosmarie Friemann[6], Cesare Indiveri[5,7], Renwick CJ Dobson[8,9], Kutti R Vinothkumar[10], Subramanian Ramaswamy[4]\***

[1]Biochemical Sciences Division, CSIR-National Chemical Laboratory, Pune, India; [2]Academy of Scientific and Innovative Research (AcSIR), Ghaziabad, India; [3]Institute for Stem Cell Science and Regenerative Medicine, Bengaluru, India; [4]Biological Sciences, Purdue University, West Lafayette, United States; [5]Department DiBEST (Biologia, Ecologia, Scienze della Terra) Unit of Biochemistry and Molecular Biotechnology, University of Calabria, Arcavacata di Rende, Italy; [6]Centre for Antibiotic Resistance Research (CARe) at University of Gothenburg, Gothenburg, Sweden; [7]CNR, Institute of Biomembranes, Bioenergetics and Molecular Biotechnologies (IBIOM), via Amendola, Bari, Italy; [8]Biomolecular Interaction Centre, Maurice Wilkins Centre for Biodiscovery, MacDiarmid Institute for Advanced Materials and Nanotechnology, and School of Biological Sciences, University of Canterbury, Christchurch, New Zealand; [9]Department of Biochemistry and Pharmacology, Bio21 Molecular Science and Biotechnology Institute, University of Melbourne, Parkville, Melbourne, Australia; [10]National Centre for Biological Sciences TIFR, GKVK Campus, Bellary Road, Bengaluru, India

**\*For correspondence:**
p.goyal@ncl.res.in (PG);
subram68@purdue.edu (SR)

**Competing interest:** The authors declare that no competing interests exist.

## eLife Assessment

This **valuable** work provides novel insights into the substrate binding mechanism of a tripartite ATP-independent periplasmic (TRAP) transporter, which may be helpful for the development of specific inhibitors. The structural analysis is **convincing**, but additional work will be required to establish the transport mechanism as well as well as binding sites for all ligands. This study will be of interest to the membrane transport and bacterial biochemistry communities.

**Abstract** *N*-Acetylneuraminic acid (Neu5Ac) is a negatively charged nine-carbon amino sugar that is often the peripheral sugar in human cell-surface glycoconjugates. Some bacteria scavenge, import, and metabolize Neu5Ac or redeploy it on their cell surfaces for immune evasion. The import of Neu5Ac by many bacteria is mediated by tripartite ATP-independent periplasmic (TRAP) transporters. We have previously reported the structures of SiaQM, a membrane-embedded component of the *Haemophilus influenzae* TRAP transport system, (Currie et al., 2024). However, none of the published structures contain Neu5Ac bound to SiaQM. This information is critical for defining the transport mechanism and for further structure-activity relationship studies. Here, we report the structures of *Fusobacterium nucleatum* SiaQM with and without Neu5Ac. Both structures are in an inward (cytoplasmic side) facing conformation. The Neu5Ac-bound structure reveals the interactions of Neu5Ac with the transporter and its relationship with the Na$^+$ binding sites. Two of the Na$^+$-binding sites are similar to those described previously. We identify a third metal-binding site that is

further away and buried in the elevator domain. Ser300 and Ser345 interact with the C1-carboxylate group of Neu5Ac. Proteoliposome-based transport assays showed that Ser300-Neu5Ac interaction is critical for transport, whereas Ser345 is dispensable. Neu5Ac primarily interacts with residues in the elevator domain of the protein, thereby supporting the elevator with an operator mechanism. The residues interacting with Neu5Ac are conserved, providing fundamental information required to design inhibitors against this class of proteins.

## Introduction

Sialic acids are nine-carbon amino sugars common on the surface of mammalian cells and on secreted molecules. Neu5Ac and *N*-glycolylneuraminic acid (Neu5Gc) are the most prevalent forms in mammals (*Lewis and Lewis, 2012*). A dynamic interplay between genetics, environmental cues, and cell signaling orchestrates the intricate regulation of sialic acid synthesis in mammals. These sugars frequently occupy terminal positions in glycolipids and glycoproteins and play pivotal roles in cell-cell interactions such as signaling, adhesion, and recognition (*Chen and Varki, 2010*). Notably, a frameshift mutation disrupts the CMP-Neu5Ac hydrolase (CMAH) gene and leads to the loss of enzymatic activity, resulting in Neu5Ac being the sole outermost sugar in humans. Conversely, primates possessing a functional CMP-Neu5Ac hydrolase, Neu5Gc is typically found as the outermost sugar (*Chou et al., 1998*).

This evolutionary divergence has profound consequences for the interactions between humans and pathogenic agents, including bacteria, parasites, and viruses (*Varki, 2009*). While commensal bacteria harness Neu5Ac as a carbon source, pathogenic bacteria, such as *Haemophilus influenzae* (*Hi*) and *Fusobacterium nucleatum* (*Fn*), evolved the ability to add Neu5Ac as the outermost sugar in their cell surface glycol-conjugates and use molecular mimicry to evade the immune system (*Bell et al., 2023*; *Severi et al., 2007*). Sialic acid is abundant in many niches [e.g. Neu5Ac] in human serum is 1.6–2.2 mM (*Sillanaukee et al., 1999*). However, the concentration of free sialic acid is often low (approximately 0.2% of the total), and much of the sialic acid is conjugated to other macromolecules on the cell surface and is, therefore, not immediately available. Secreted sialidases from bacteria cleave Neu5Ac from mucins and other biomolecules within their niche (*Sillanaukee et al., 1999*). Cleaved Neu5Ac is transported into the periplasm of Gram-negative bacteria via porin-like $\beta$−barrel proteins. NanC from *E. coli* is the best-studied porin specific for sialic acids (*Wirth et al., 2009*).

Our groups have made significant strides in elucidating the structural intricacies of the proteins involved in sequestration, uptake, catabolism, and incorporation of Neu5Ac by bacteria (*Bose et al., 2019*; *Coombes et al., 2020*; *Currie et al., 2021*; *Davies et al., 2019*; *Horne et al., 2020*; *Kumar et al., 2018*; *Manjunath et al., 2018*; *North et al., 2016*; *Gangi Setty et al., 2018*). Bacteria rely on specialized Neu5Ac transporters within their cytosolic membranes, and four distinct classes of Neu5Ac transporters have been identified: sodium solute symporters (SSS) (*North et al., 2018*), major facilitator superfamily (MFS), ATP-binding cassette (ABC), and TRAP transporters (*Davies et al., 2023*; *Peter et al., 2022*). TRAP transporters that manifest as two- or three-component systems consisting of a substrate-binding protein (SiaP) and transmembrane protein (SiaQM) (*Rosa et al., 2018*) are of particular interest due to their absence in eukaryotes. Deleting the sialic acid transporter abolishes Neu5Ac uptake, rendering bacteria incapable of incorporating Neu5Ac into lipopolysaccharides. Without a SiaQM transporter, bacteria form defective biofilms, have a lower cell density, and experience higher cell death (*Allen et al., 2005*). Additionally, amino acids, C4-dicarboxylates, aromatic substrates, and alpha-keto acids are also transported by TRAP transporters (*Vetting et al., 2015*).

The first structures of SiaP were obtained from *H. influenzae* in an unliganded form and bound to Neu5Ac (*Johnston et al., 2008*; *Müller et al., 2006*). The binding of Neu5Ac to SiaP results in the closure of the two domains of the protein and the mechanism has been described as a 'Venus fly trap' mechanism, a common phenomenon observed in many ligand-binding proteins (*Felder et al., 1999*). Structural and thermodynamic analyses of SiaP from *F. nucleatum*, *V. cholerae*, and *P. multocida* have revealed a conserved binding site, dissociation constant of Neu5Ac in the nanomolar range, and enthalpically driven substrate binding (*Gangi Setty et al., 2018*). Using smFRET on *Vc*SiaP, Peter and co-workers showed that conformational switching is strictly substrate-induced and that binding of the substrate stabilizes the interactions between the two domains (*Peter et al., 2021*). All the SiaP structures show the presence of a conserved Arginine that binds to the C1-carboxylate of Neu5Ac, and

this Arg residue is critical as the high electrostatic affinity may be important to have a strong binding affinity that sequesters the small amounts that reach the bacterial periplasmic space (*Glaenzer et al., 2017*).

The second component of the TRAP transporter is the transmembrane protein SiaQM, which is characterized by two distinct domains: the Q-domain and M-domain. Two SiaQM transporter structures have been reported by cryo-EM, one from *H. influenzae* (*Hi*SiaQM) ranging from 2.95 to 4.70 Å resolution and another from *Protobacterium profundum* (*Pp*SiaQM) at 2.97 Å resolution (*Currie et al., 2024*; *Davies et al., 2023*; *Peter et al., 2022*).

The first structure of *Hi*SiaQM (4.7 Å resolution) demonstrated that it is composed of 15 transmembrane helices (TM) and two helical hairpins. *Hi*SiaQM likely functions as a monomeric unit, although a dimeric form has recently been reported (*Currie et al., 2024*; *Peter et al., 2022*). Similarly, *Pp*SiaQM is composed of 16 TM helices (*Davies et al., 2023*). Based on these structures, an elevator-type mechanism has been proposed for TRAP transporters to move substrates from the periplasm to the cytoplasm (*Peter et al., 2024*). Apart from the dimeric HiSiaQM structure, the other two structures determined used a megabody bound to SiaQM for cryo-EM analysis. In both cases, the megabody was bound to the extracellular side of the QM complex, featuring a deep open cavity on the cytosolic side (*Figure 2—figure supplement 2*). This observation suggested that the transporter was captured in an inward-facing conformation, although the higher resolution dimeric HiSiaQM structure, even without a fiducial megabody, is also in the inward-facing conformation. Notably, the transport of Neu5Ac by TRAP transporters requires at least two sodium ions (*Currie et al., 2024*; *Davies et al., 2023*; *Mulligan et al., 2012*; *Mulligan et al., 2009*).

In sialic acid-rich environments such as the gut and saliva, bacterial virulence often correlates with their capacity to utilize Neu5Ac as a carbon source (*Almagro-Moreno and Boyd, 2010*; *Corfield, 2015*; *Haines-Menges et al., 2015*). Although inhibitors targeting viral sialidases (neuraminidases) have been developed as antiviral agents (e.g. oseltamivir, zanamivir, and peramivir) (*Glanz et al., 2018*), no drug currently targets bacterial infections by inhibiting sialic acid sequestration, uptake, catabolism, or incorporation. Human analogs of TRAP transporters are notably absent, underscoring their potential as promising therapeutic targets for combatting bacterial infections.

Both *Hi*SiaQM and *Pp*SiaQM structures lack information on the Neu5Ac binding site, which was identified based on modeling studies that relied on the ligand-bound structure of *Vc*INDY (*Kinz-Thompson et al., 2022*). Moreover, only the structures of SiaQM in the elevator-down conformation (inward-facing) have been reported, and further conformations along the transport cycle remains to be elucidated. The conformation of Neu5Ac bound to the transport domain may provide clues as to how it is received from the substrate-binding protein. In this study, we present the cryo-EM structures of SiaQM from *F. nucleatum*, both in its unliganded form and Neu5Ac bound form. The unliganded structure has density for two $Na^+$-binding sites, whereas the liganded form has density for two $Na^+$-binding sites and an additional metal-binding site.

## Results and discussion
### Construct design, isolation, and strategy for structural determination

The TRAP transporter from *F. nucleatum* (*Fn*SiaQM) was tagged with a green fluorescent protein (GFP) at the C-terminus and expressed in BL21 (DE3) cells for protein expression and detergent stabilization trials. Detergent scouting was performed to identify the preferred detergent for *Fn*SiaQM purification using fluorescence detection size-exclusion chromatography (*Kawate and Gouaux, 2006*). n-Dodecyl-β-D-maltopyranoside (DDM) was chosen as it solubilized and stabilized the protein better than other detergents. *Fn*SiaQM was re-cloned into the pBAD vector with N-terminal 6 X -histidine followed by Strep-tag II for large-scale purification and expressed in TOP10 (Invitrogen) cells. The purified *Fn*SiaQM protein in the detergent micelles eluted as a monodisperse peak in size exclusion chromatography (*Figure 1—figure supplement 1*). Subsequently, *Fn*SiaQM was further reconstituted in MSP1D1 nanodiscs, along with *E. coli* polar lipids.

Initially, we attempted to determine the *Fn*SiaQM structure using X-ray crystallography. After failing to obtain well-diffracting crystals, we switched to single-particle electron cryo-microscopy (cryo-EM). To achieve a reasonable size and better particle alignment, we raised nanobodies against *Fn*SiaQM in Alpaca (Center for Molecular Medicine, University of Kentucky College of Medicine). Two high-affinity

binders (T4 and T7) were identified and tested for complex formation using *Fn*SiaQM. Hexa-histidine-tag-based affinity chromatography was used to purify the nanobody from the host bacterial periplasm, producing a monodisperse peak in the size-exclusion chromatogram. Both nanobodies were added to *Fn*SiaQM in DDM micelles and tested for complex formation. Although both nanobodies bound to the transporter, the T4 nanobody was selected over T7 because of its higher expression. The structures of the purified *Fn*SiaQM-nanobody (T4) complexes in nanodiscs with and without Neu5Ac bound were determined using cryo-EM.

## Structure of the FnSiaQM-nanobody complex

The nanodisc reconstituted *Fn*SiaQM, and the bound nanobody is a monomeric complex in the cryo-EM structure. The overall global resolutions of both the unliganded *Fn*SiaQM-nanobody complex and the Neu5Ac bound form is ~3.2 Å. The maps were of good quality and allowed us to build all the TM helices of FnSiaQM unambiguously and both the unliganded and the Neu5Ac bound form clearly allow us to trace the protein and the nanobody (*Figure 1A, B, C and D*). The best resolution for both structures are observed in the interior of the SiaQM protein (*Figure 1—figure supplement 2* and *Figure 1—figure supplement 3*). Owing to their flexibility, the N-terminal 6 X histidine and Strep II tags on *Fn*SiaQM-nanobody complex were not visible in the cryo-EM maps. Similarly, the last few amino acids at the C-terminus were not built because of poor density; the rest of the helices showed well-defined density (*Figure 1—figure supplement 4*). The density of the nanodisc was visible but of insufficient quality for model building. The density of the bound Neu5Ac is clear and in the higher-resolution region. The observed density around the ligand binding site for the two maps clearly indicate minimal structural changes around the binding pocket (*Figure 1C and D*, inset). The details of the structure quality and refinement parameters are shown in *Table 1*, and a flow diagram for structure determination and resolution statistics is shown in *Figure 1—figure supplement 5A and B*.

As the name suggests, many TRAP transporters comprise three units: a substrate-binding protein (SiaP) and two membrane-embedded transporter units (SiaQ and SiaM) (*Severi et al., 2007*). In *Fn*SiaQM, the two transporter units are fused by a long connecting helix, similar to other TRAP transporters such as *H. influenzae* TRAP (*Hi*SiaQM) (*Currie et al., 2024*; *Peter et al., 2022*). The Q-domain of *Fn*SiaQM consists of the first four long helices, which are tilted in the membrane plane. This tilting creates a large contact surface area between the Q- and M- domains, which is dominated by buried hydrophobic residues. A small connecting helix lies perpendicular to the cell membrane plane and does not contact either domain (*Figure 2*). The lack of interaction between this connecting helix and the two domains suggests that it may be redundant for the transporter function. For example, *Pp*SiaQM contains two separate polypeptides and does not require a connecting helix for its function (*Davies et al., 2023*).

While there is no direct observation, the possible mode of function of the transporter suggests that the N-terminus of *Fn*SiaQM lies on the cytoplasmic side, whereas its C-terminus lies on the periplasm side of the membrane. The nanobody is bound to the periplasmic side of *Fn*SiaQM, making no contact with the cytoplasmic side of the protein (*Figure 1A and B*). The M-domain is located towards the C-terminus and comprises the bulk of the transporter. It consists of 10 transmembrane helices with two hairpins that do not cross the entire membrane.

Superposition of the reported structures from *Hi*SiaQM and *Pp*SiaQM demonstrate that all are in inward (cytoplasmic) open and elevator-down conformations (*Figure 2—figure supplement 2*). There are two conserved Na$^+$ ion binding sites in these proteins (*Currie et al., 2024*; *Davies et al., 2023*; *Peter et al., 2022*). Structurally, the M-domain can be divided into an outer variable scaffold domain and a centrally conserved transport domain. The scaffold domain is composed of six TM helices and serves as a support for the transport domain. It also interacts with the Q-domain to form a structurally rigid domain. The Q and scaffold domains are the least conserved among the SiaQM domains, suggesting that they are not directly involved in substrate transport (*Figure 2*, *Figure 2—figure supplement 3*).

To determine whether substrate binding resulted in conformational changes, we solved the structure of the ligand-bound form of *Fn*SiaQM. The superposition of the two structures revealed a high degree of similarity with an RMSD of 0.3 Å over all Cα atoms (*Figure 2A*). A comparison of the binding site residues revealed that they are in a similar conformation. The transporter has a large cytoplasmic facing open cavity, suggesting that the unliganded as well as Neu5Ac bound structures are

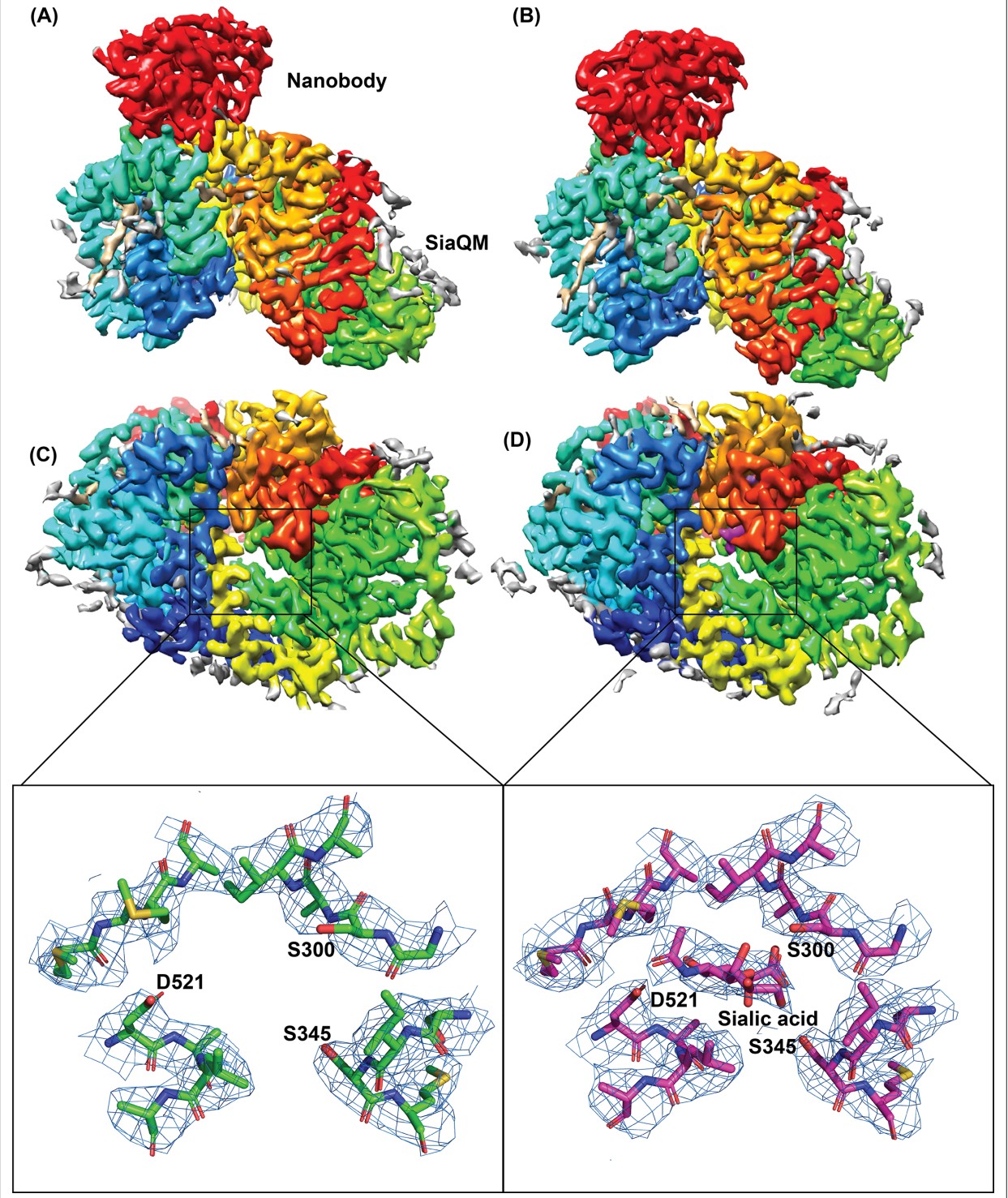

**Figure 1.** Architecture of Fusobacterium nucleatum (FnSiaQM) with nanobody. (**A and B**) Cryo-EM maps of FnSiaQM unliganded and N-Acetylneuraminic acid (Neu5Ac) bound at 3.2 and 3.17 Å, respectively. The TM domain of FnSiaQM is colored using the rainbow model (N-terminus in blue and C-terminus in red). The nanobody density is colored in red. The density for modeled lipids is colored in tan and the unmodeled density in gray. The figures were made with Chimera at thresholds of 1.2 and 1.3 for the unliganded and Neu5Ac-bound maps. (**C and D**) The cytoplasmic view of apo and Neu5Ac bound FnSiaQM, respectively. Color coding is the same as in panels A and B. The density corresponding to Neu5Ac and sodium ions are in purple. The substrate binding sites of apo and Neu5Ac bound FnSiaQM are shown with key residues labeled. The density (blue mesh) around these atoms was made in Pymol with 2 and 1.5 σ for the apo and the Neu5Ac structures, respectively, with a carve radius of 2 Å.

*Figure 1 continued on next page*

*Figure 1 continued*

The online version of this article includes the following source data and figure supplement(s) for figure 1:

**Figure supplement 1.** Size exclusion chromatography of Tripartite ATP-independent periplasmic (TRAP) transporter from *Fusobacterium nucleatum* (FnSiaQM).

**Figure supplement 1—source data 1.** Raw, uncropped image for gel shown in *Figure 1—figure supplement 1*.

**Figure supplement 1—source data 2.** Uncropped, labelled image for gel shown in *Figure 1—figure supplement 1*.

**Figure supplement 2.** The gray surface represents the areas of the density that have been modeled in the unliganded (**A**) and the liganded (**B**) forms.

**Figure supplement 3.** The local resolution maps of the unliganded and the liganded *Fusobacterium nucleatum* (*Fn*SiaQM) respectively.

**Figure supplement 4.** Density maps of the unliganded (**A**) and liganded (**B**) FnTRAP helices.

**Figure supplement 5.** Cryo-EM workflow and analysis of the *Fusobacterium nucleatum* (*Fn*SiaQM) protein.

in an inward-open conformations. The nomenclature used for labeling the secondary structures uses a description of the published *Hi*SiaQM structure with similar architecture (*Currie et al., 2024*). The elevator part of the domain has one helix-loop-helix motif called HP1 (HPin), which is directed towards the cytoplasmic side. A structurally homologous helix-loop-helix domain, HP2 (HPout), is present on the periplasmic side. The arrangement of TM helices have been described in detail in *Hi*SiaQM and *Pp*SiaQM structures (*Davies et al., 2023*; *Peter et al., 2022*). Interestingly, this domain has twofold inverted symmetry, as found in other TRAP transporters. The entire domain is populated with highly conserved residues, suggesting that it may play a direct role in substrate transport (*Figure 2—figure supplement 3*). A single molecule of Neu5Ac is bound to the aperture formed by HP1 and HP2 in the central core transport domain (*Figure 2B*). The Neu5Ac binding site has a large solvent-exposed vestibule towards the cytoplasmic side, while its periplasmic side is sealed off. Cryo-EM map shows the presence of multiple densities that could be modeled as lipids, possibly preventing the substrate from leaving the transporter. However, the densities are not well defined to model them as specific lipids, hence they have not been modeled. We describe this as the 'inward-facing open state' with the substrate-bound. While, both the HP1 and HP2 loops have been hypothesized to be involved in gating, in the human neutral amino acid transporter (ASCT2), (which also uses the elevator mechanism), only the HP2 loops have been shown to undergo conformational changes to enable substrate binding and release (*Garaeva et al., 2019*). Hence, it is suggested that there is a single gate that controls substrate binding. Superposition of the *Pp*SiaQM and *Hi*SiaQM structures do not reveal any change in these loop structures upon substrate binding. For TRAP transporters, the substrate is delivered to the QM protein by the P protein; hence, these loop changes may not play a role in ligand binding or release. This may support the idea that there is minimal substrate specificity within SiaQM and that it will transport the cargo delivered by SiaP, which is more selective.

## Sodium ion binding site

The transport of molecules across the membrane by TRAP transporters depends on $Na^+$ transport (*Currie et al., 2024*; *Davies et al., 2023*; *Mulligan et al., 2012*; *Mulligan et al., 2009*). We observed cryo-EM density at conserved $Na^+$ binding sites and modeled two $Na^+$ ions in the unliganded- and ligand-bound forms (*Figure 3*). The two sites that are present in the unliganded form are also conserved in other reported SiaQM structures *Currie et al., 2024*; *Davies et al., 2023*; (*Figure 2B*, *Figure 2—figure supplement 4*), compared with Figure 5 in *Currie et al., 2024*. The Na1 site interacts with residues at the HP1 site and helix 5 (*Figure 3A*, *Figure 3—figure supplement 1*). The Na2 site combines residues at the HP2 site and a short stretch that splits helix 11 into two parts (*Figure 3A*, *Figure 3—figure supplement 1B*). These two sites have been previously well described (*Currie et al., 2024*). These two sodium ion binding sites are also conserved in the structure of VcINDY (*Figure 3—figure supplement 2*; *Sauer et al., 2022*). In both cases, the sodium ions are bound at the helix-loop-helix ends of HP1 and HP2. The binding sites utilize both side chains and main chain carbonyl groups. The number of main chain carbonyl interactions suggests that they are critical, and using main chain rather than side chain interactions minimizes the likelihood of point mutations affecting the binding.

Surprisingly, we observed density for an additional metal binding site. This site is distinctly located towards the cytoplasmic side of the transporter and away from the Neu5Ac binding site. It interacts with helices that form HP1 and the next helix 5 a (*Figure 3A*). Interestingly, a similar site was observed in the SSS Neu5Ac transporter, in which the third site was located away from the other

**Table 1.** Details of cryo-EM data collection, processing, refinement, and built model.

| Data collection and processing | Unliganded form | Neu5Ac-bound |
|---|---|---|
| EMDB | 38926 | 38925 |
| PDB | 8Y4X | 8Y4W |
| Magnification | 75000 | 75000 |
| Voltage (kV) | 300 | 300 |
| Electron dose (e⁻/Å³) | 27.7 | 24.67 |
| Defocus Range (nm) | 1800–3000 | 1600–2800 |
| Pixel size (Å) | 1.07 | 1.07 |
| Symmetry imposed | C1 | C1 |
| Initial particle images (no.) | 385668 | 653554 |
| Final particle images (no.) | 141272 | 225006 |
| Map resolution (Å) | 3.20 | 3.16 |
| FSC threshold | 0.143 | 0.143 |
| Map resolution range (Å) | 2.7–5.5 | 2.6–4.1 |
| **Refinement** | | |
| Model composition | | |
| Non-hydrogen atoms | 5801 | 5873 |
| Protein residues | 732 | 732 |
| Ligand | 2 Na⁺;1 Lipid | 3 Na⁺; 2 Lipid; 1 Neu5Ac |
| B-factor (Å²) | | |
| Protein (mean) | 46.72 | 43.60 |
| Ligand (mean) | 45.88 | 56.25 |
| R.m.s deviations | | |
| Bond lengths (Å) | 0.005 | 0.004 |
| Bond angles (°) | 0.531 | 0.512 |
| Validation | | |
| MolProbity score | 1.76 | 1.54 |
| Clashscore | 5.57 | 5.07 |
| CaBALM outliers (%) | 1.52 | 1.52 |
| Rotamer outliers (%) | 1.77 | 1.13 |
| Ramachandran plot (%) (Favored/Allowed/Disallowed) | (96.02/3.98/0) | (96.43/3.57/0) |

Na⁺ binding sites (*Wahlgren et al., 2018*). Peter et al., proposed that Asp304 coordinates with this Na⁺ ion, showing loss-of-function when mutated to an alanine residue (*Peter et al., 2022*). While it is tempting to label it as a third Na⁺-binding site, the metal-ligand distances are longer (*Figure 3— figure supplement 1C*). Hence, we conservatively designated this site as a metal-binding site. The coordinated movement of different Na⁺ ions is expected to provide the energy to create conformational changes that lead to the movement of Neu5Ac from the periplasm to the cytoplasm. While we found two Na⁺-binding sites and a third metal-binding site, it is not clear how the movement of these sites can cause the conformational changes required for the motion of the elevator along with the ligand. It is also unclear what the function of the third metal-binding site is beyond the mutagenesis data, which suggests that the lack of this site results in a loss of function.

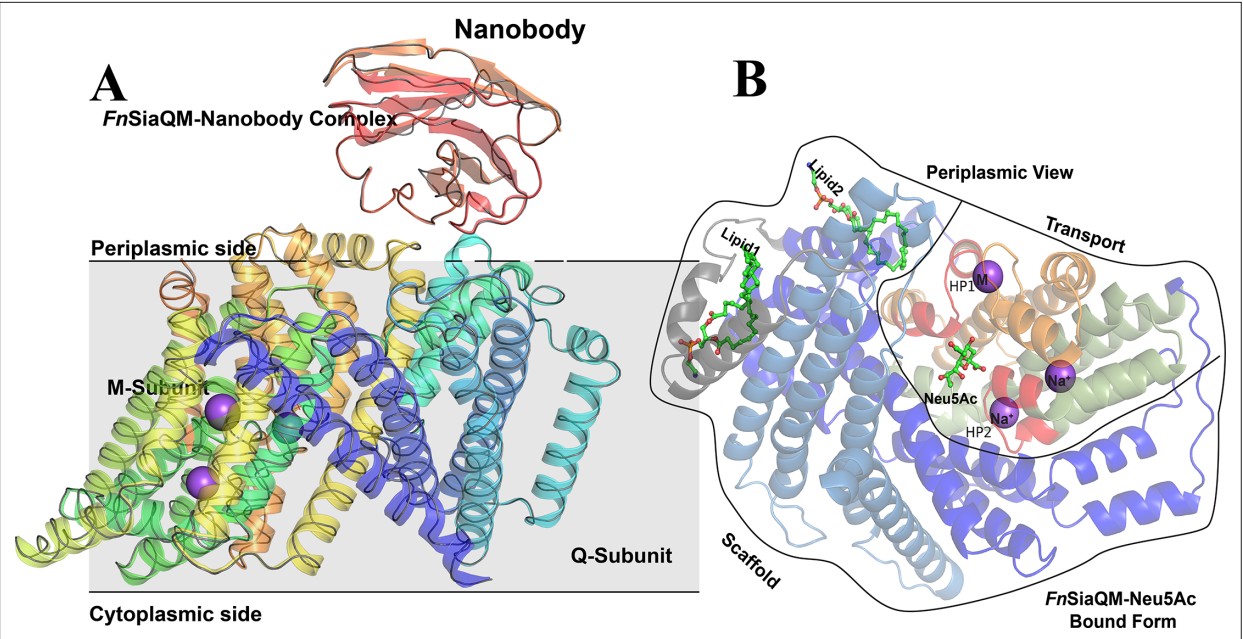

**Figure 2.** *Fusobacterium nucleatum* (*Fn*SiaQM)-Nanobody complex with N-Acetylneuraminic acid (Neu5Ac) bound form. (**A**) A cartoon representation of the *Fn*SiaQM-Nanobody complex structure. The SiaQM polypeptide has been colored in rainbow with the N-terminus starting in blue and the C-terminus ends in orange. The nanobody is shown in cartoon representation and in red color. Two of the modeled sodium sites are shown as purple spheres. A ribbon representation of the unliganded structure of *FnSiaQM* is superposed in gray. The superposition reveals that the overall structures are similar. (**B**) Cartoon representation of the *Fn*SiaQM structure bound to Neu5Ac. In shades of blue are the helices that form the scaffold domain. In gray is the connecting domain. In olive, orange, and red is the elevator domain. HP1 and HP2, the two helix-loop-helix motifs, are in red and marked. The positions of the known Na$^+$ ions and metal ion (M) are in spheres, and the position of Neu5Ac is shown in ball and stick.

The online version of this article includes the following figure supplement(s) for figure 2:

**Figure supplement 1.** Lipid binding to FnSiaQM and Electrostatic potential map.

**Figure supplement 2.** Superposition of SiaQM structure from *F.nucleatum*, *H. influenzae* and *P. profundum* with respective bound nanobody or megabody.

**Figure supplement 3.** SiaQM protein sequence alignment.

**Figure supplement 4.** Ribbon diagram showing the superposition of the *Hi*SiaQM, *Pp*SiaQM, and *Fn*SiaQM (liganded and unliganded) structures.

## Lipid binding to FnSiaQM

We have modeled two phosphatidylethanolamine (PE) lipids in the maps of unliganded *Fn*SiaQM (*Figure 2—figure supplement 1A and B*). The *Fn*SiaQM nanodisc reconstitution procedure included the addition of *E. coli* polar lipids, where PE is the most abundant lipid. One of the lipids binds between the connector helix and the Q-domain (*Figure 2B*, **Lipid1**) and is present in both the unliganded and Neu5Ac bound structure of *Fn*SiaQM. This lipid is present at a similar position in the *Hi*SiaQM structure (*Currie et al., 2024*). However, we observe a second lipid molecule (*Figure 2B*, **Lipid2**) that is bound between the stator and elevator domains only in the Neu5Ac bound structure. It is tempting to hypothesize that this lipid molecule is displaced during the movement of the elevator domain; however, this requires further investigation. A similar observation was also made in the transporter GltPh that invokes the elevator mechanism (*Wang and Boudker, 2020*).

## Activity of FnSiaQM

To demonstrate that the purified protein is active and transports Neu5Ac across the membrane, we performed transport assays. *Fn*SiaQM was reconstituted in proteoliposomes containing intraliposomal potassium. Valinomycin was then incorporated into these proteoliposomes to establish an inner-negative membrane potential. Periplasmic binding protein (*Fn*SiaP) and radioactive Neu5Ac were added to the proteoliposomes to initiate transport (*Figure 4A*). In this experimental setup, a significant accumulation of radiolabeled Neu5Ac was measured in the presence of extraliposomal Na$^+$-gluconate when the artificial membrane potential was imposed by the addition of valinomycin

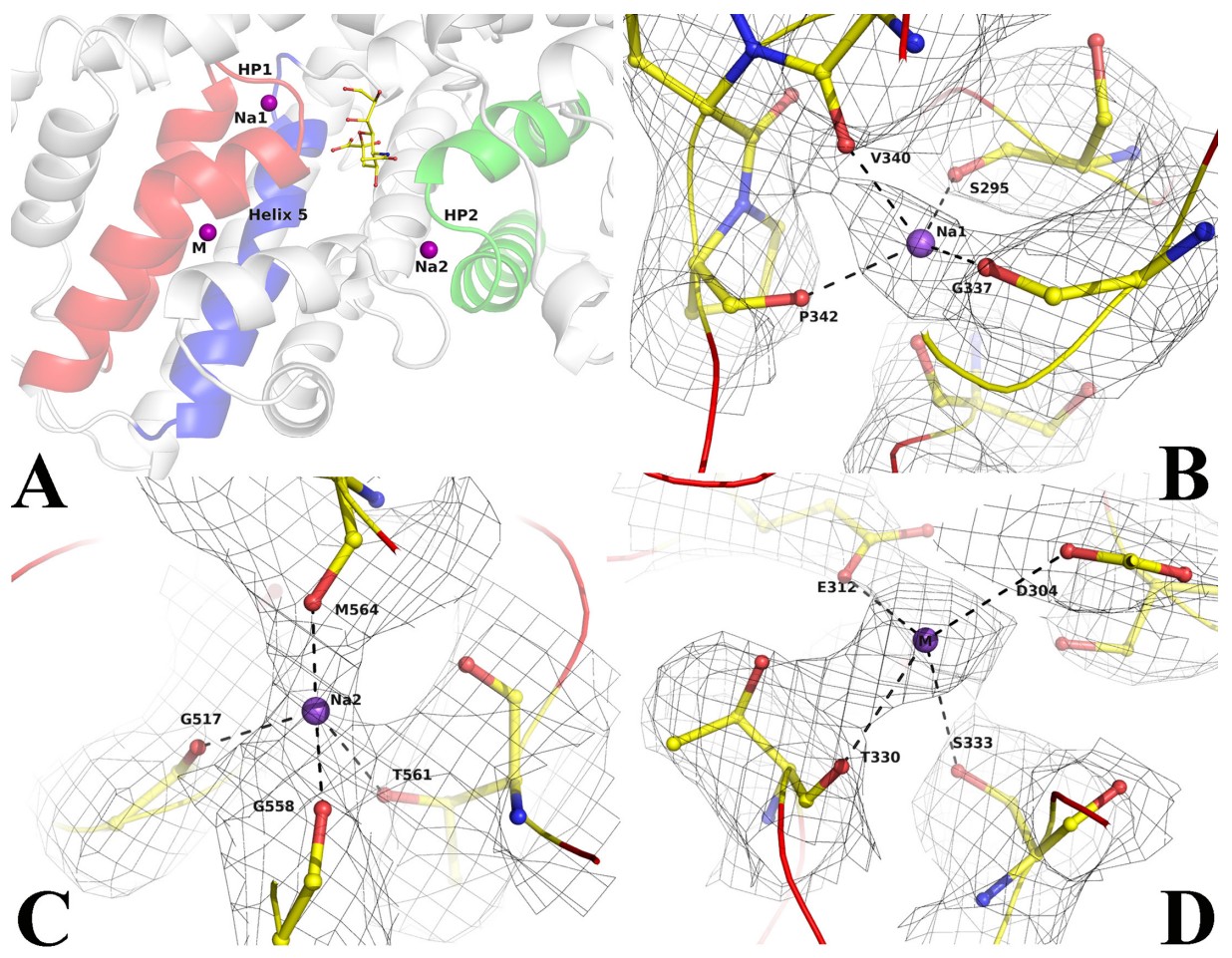

**Figure 3.** Ion binding sites in *Fusobacterium nucleatum* (*Fn*SiaQM). (**A**) Close-up view of the N-Acetylneuraminic acid (Neu5Ac) and the two Na$^+$ and metal binding sites. In red is the HP1 helix-loop-helix. In green is the HP2 helix-loop-helix. In blue is helix 5 A, the purple spheres are the two Na$^+$ ion binding sites and metal binding site (M), and the bound Neu5Ac is shown in ball and stick. (**B–D**) Density and interaction details of the Na1, Na2, and M sites, respectively. The figures are made with a contour of 1.0 r.m.s. in PyMol.

The online version of this article includes the following figure supplement(s) for figure 3:

**Figure supplement 1.** Metal ion co-ordination to FnSiaQM.

**Figure supplement 2.** An overlay of sodium ions binding helix-loop-helix regions.

(*Figure 4B*, **black squares**). When ethanol was used as a control for valinomycin, a much lower accumulation of radiolabeled Neu5Ac was observed (*Figure 4B*, **open squares**). Importantly, transport was negligible when Na$^+$ was absent in the extra-liposomal environment or when K$^+$ was absent in the intraliposomal environment. Transport was negligible when soluble *Fn*SiaP was omitted from the assay, further demonstrating the requirement of periplasmic binding proteins (*Figure 4B*, **stars**). The energetics of transport is similar to that of other SiaPQM that have been characterized (*Mulligan et al., 2012*).

### Architecture of Neu5Ac binding site

While the structures of the unliganded and Neu5Ac bound *Fn*SiaQM are in similar inward-open conformation, the density of Neu5Ac is unambiguous in the liganded form (*Figure 1D* (**inset**), **5** A). The overall architecture of the Neu5Ac binding site is similar to that of citrate/malate/fumarate in the di/tricarboxylate transporter of *V. cholerae* (*Vc*INDY), but the residues involved in providing specificity are different (*Kinz-Thompson et al., 2022*; *Mancusso et al., 2012*; *Nie et al., 2017*; *Sauer et al., 2022*). Neu5Ac binds to the transport domain without direct interactions with the residues in the scaffold domain. The majority of the interactions are with residues in the HP1 and HP2 loops of

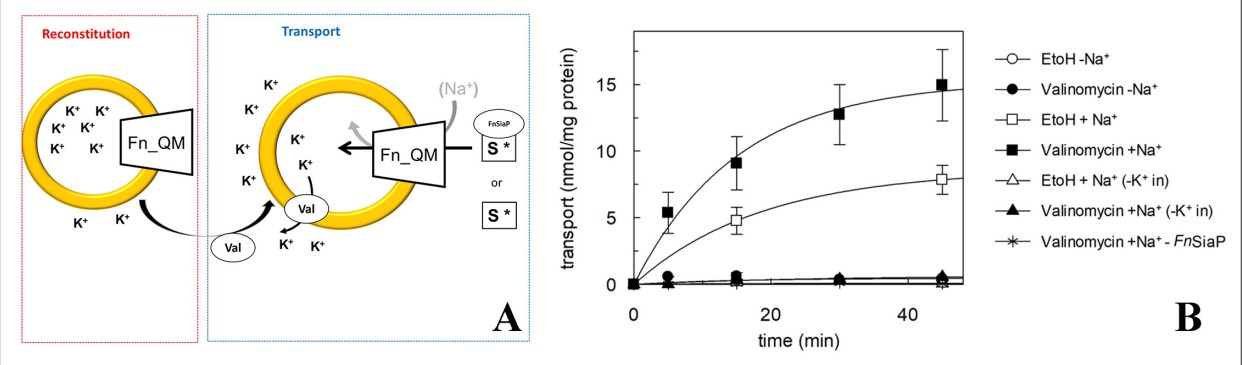

**Figure 4.** Proteoliposome transporter assays for *Fusobacterium nucleatum* (*Fn*SiaQM). (**A**) Schematic diagram showing the experimental setup. First, *Fn*SiaQM is incorporated into proteoliposome in the presence of internal K⁺. Then, valinomycin (val) is added to induce the efflux of K⁺ down its concentration gradient, imposing an artificial membrane potential. To start transport measurement, *Fn*SiaP, [³H]-Neu5Ac, and Na⁺-gluconate are added in the extraliposomal environment. (**B**) Time course of N-Acetylneuraminic acid (Neu5Ac) uptake into proteoliposomes reconstituted with *Fn*SiaQM. In black circle, black square, white triangles, black triangle, asterisk, are with conditions, where valinomycin was added to facilitate K⁺ movement before transport. Ethanol was added instead of valinomycin as a control in the white circle, white square, and white triangle. In white square, black square, white triangle, black triangle, 10 mM Na⁺-gluconate was added together with 5 µM [³H]-Neu5Ac and 0.5 µM *Fn*SiaP; in white triangle and black triangle proteoliposomes are prepared without internal K⁺; in black asterisk transport is measured in the presence of 10 mM Na⁺-gluconate, 5 µM [³H]-Neu5Ac and in the absence of *Fn*SiaP. Uptake data were fitted in a first-order rate equation for time course plots. Data are means±s.d. of three independent experiments.

the transport domain (**Figure 5B**). Asp521 (HP2), Ser300 (HP1), and Ser345 (helix 5) interact with the substrate through their side chains, except for one interaction between the main chain amino group of residue 301 and the C1-carboxylate oxygen of Neu5Ac. Mutation of the residue equivalent to Asp521 has been shown to result in loss of transport (**Peter et al., 2022**). To evaluate the role of residues Ser-300 and Ser-345, we mutated them to alanine and performed the transport assays.

The data clearly showed that the Ser300Ala mutant was inactive, whereas the Ser345 mutation did not affect *Fn*SiaQM functionality (**Figure 5C**). This suggests that the interaction of Neu5Ac with the residues in HP1 and HP2 is critical for transport. The carboxylate oxygens of the C1 atom of Neu5Ac interact with Ser300 and Ser345 and the main chain of Ala301 (**Figure 5B**). Ser 345 OG is 3.5 Å away from the C1-carboxylate oxygen – a distance that would result in a weak interaction between the two groups. It is, therefore, not surprising that the mutation into Ala did not affect transport. The space created by the mutation can be occupied by a water molecule. The N5 atom of the *N*-acetyl group of the sugar interacts with Asp521. These interactions are conserved even if Neu5Ac is converted into Neu5Gc, as the addition of an extra hydroxyl group at the C11 position does not break this interaction, and there is sufficient cavity space for modifications in C11 (**Figure 5D**). Cavity Plus (**Wang et al., 2023**) estimated the cavity to be 1875 Å³ with a druggability score of 3631, suggesting that the environment of the binding site is highly suited for drug binding (**Wang et al., 2023**).

No direct measurement of $K_d$ of Neu5Ac binding to the SiaQM region of the transporter is available. We infer from the interactions it makes and the fact that high concentrations of Neu5Ac are required to obtain a complex (30 mM), the affinity is unlikely to be high. Neu5Ac binds with nanomolar affinity ($K_d = 45$ nM) to *Fn*SiaP (**Figure 6A**) and several electrostatic interactions stabilize this binding (**Gangi Setty et al., 2014**). The binding of Neu5Ac to the SSS-type transporter, which also transports Neu5Ac across the cytoplasmic membrane, was also stabilized by several electrostatic interactions (**Figure 6B**). The polar groups bind to both the C1-caboxylate side of the molecule and the C8-C9 carbonyls, suggesting that *Proteus mirabilis* Neu5Ac transporter (SSS type) evolved specifically to transport nine-carbon sugars such as Neu5Ac (**Wahlgren et al., 2018**). Interestingly, even the dicarboxylate transporter from *V. cholerae* (*Vc*INDY) binds to its ligand via electrostatic interactions with both carboxylate groups (**Figure 6C**; **Kinz-Thompson et al., 2022**; **Mancusso et al., 2012**; **Nie et al., 2017**; **Sauer et al., 2022**). The high affinity of the substrate-binding component (*Fn*SiaP) to Neu5Ac is physiologically relevant because it sequesters Neu5Ac in a volume where the concentration of Neu5Ac is very low. In contrast, as Neu5Ac is delivered to the SiaQM component by the SiaP component, the affinity could be lower, an argument also made by Peter *et. al.* in their recent work

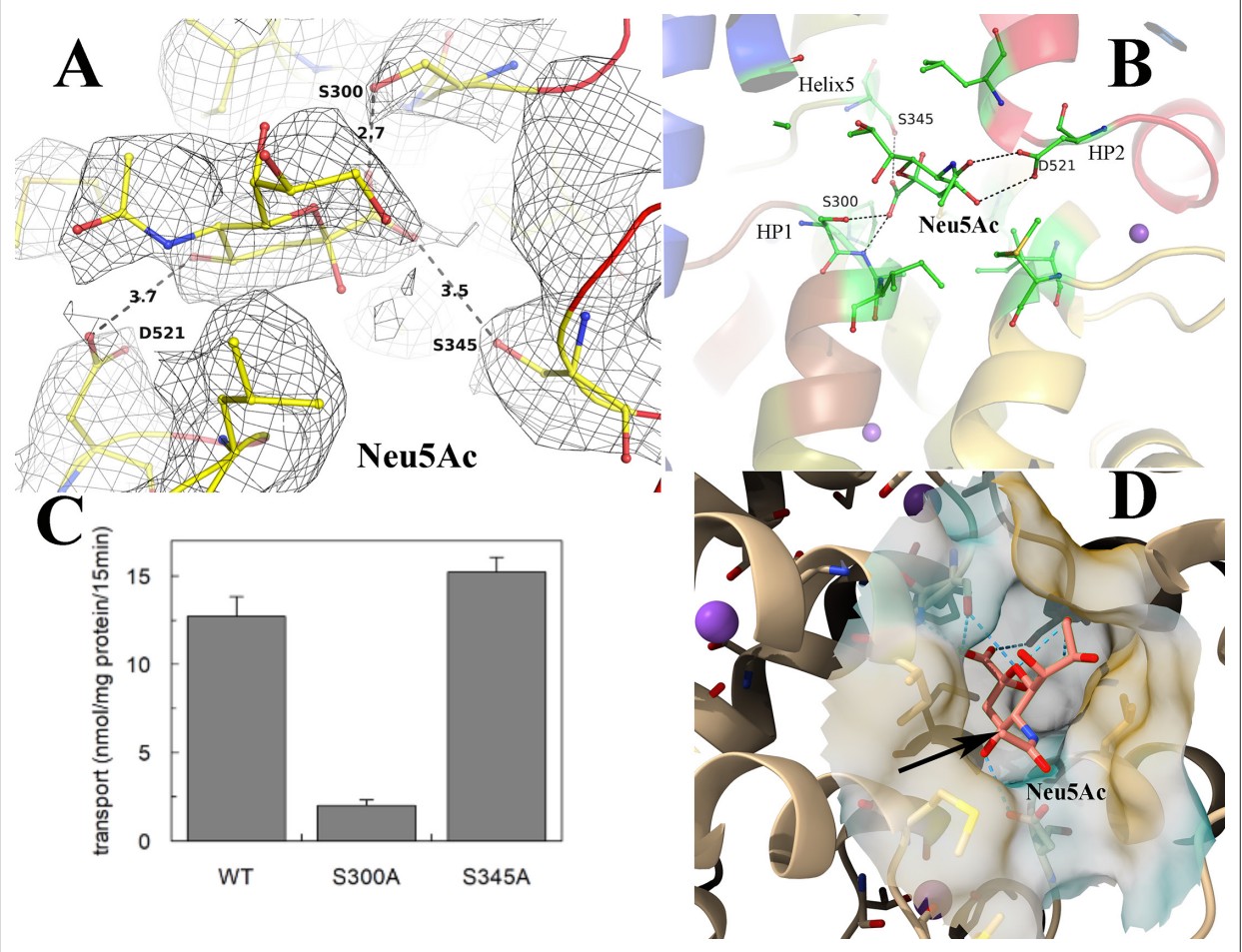

**Figure 5.** Validation of N-Acetylneuraminic acid (Neu5Ac) binding pocket in Fusobacterium nucleatum (*Fn*SiaQM) transporter. (**A**) The fit of the modeled Neu5Ac into the density, contoured to 0.9 * r.m.s. The figure also shows the fit to the density of the residues that interact with Neu5Ac and the distances to key residues discussed in the manuscript. (**B**) The interactions of Neu5Ac with side chains of interacting residues. Ser300 Oγ is 2.8 Å from the C1-carboxylate oxygen of Neu5Ac, while the main chain NH of residue 301 is 2.6 Å away. The other close polar side chain is that of Ser345γ, which is 3.3 Å away. The Neu5Ac O10 is 2.9 Å from Asp521 Oδ. (**C**) Transport of Neu5Ac in 15 min. Transport was started by adding 5 µM [³H]-Neu5Ac, 10 mM Na⁺-gluconate, and 0.5 µM *Fn*SiaP to proteoliposomes in the presence of valinomycin (expected membrane potential, Δ Ψ, −117.1 mV). Data are expressed as nmol/mg prot/15 min ±s.d. of three independent experiments. (**D**) The cavity of Neu5Ac is exposed from the cytoplasmic side. The cavity is large and extends into the cytoplasmic side. The black arrow shows the methyl group with an extra hydroxyl in Neu5Gc.

(***Peter et al., 2024***). The corollary of this argument is that the specificity of the transport system (or the choice of molecules that can be transported) is likely to be determined by the substrate-binding component. There is probably very little selectivity in the SiaQM component, which is also reflected by fewer interactions.

## Conclusion

This is the first report of the structure of a SiaQM from the TRAP-type transporter of Gram-negative bacteria with bound Neu5Ac. The structure shows no direct interaction between Neu5Ac and Na⁺ ions. The affinity of Neu5Ac for periplasmicSiaP is in the nanomolar range, with many polar interactions ($K_d$ = 45 nM) (***Gangi Setty et al., 2014***). However, relatively fewer polar interactions stabilize Neu5Ac binding to the open binding pocket of SiaQM, and the affinity is probably poor. We used APBS to calculate the electrostatic charge distribution on SiaQM (***Figure 2—figure supplement 1C***; ***Jurrus et al., 2018***). A view of the structure of Neu5Ac bound SiaQM from the periplasmic side towards the bound Neu5Ac shows that the sugar is bound in the cavity, the entrance of which is repulsive to the binding of a negatively charged sugar like Neu5Ac. This likely prevents Neu5Ac from binding to the SiaQM from the cytoplasmic side. The structure of the protein in the outward-facing conformation is

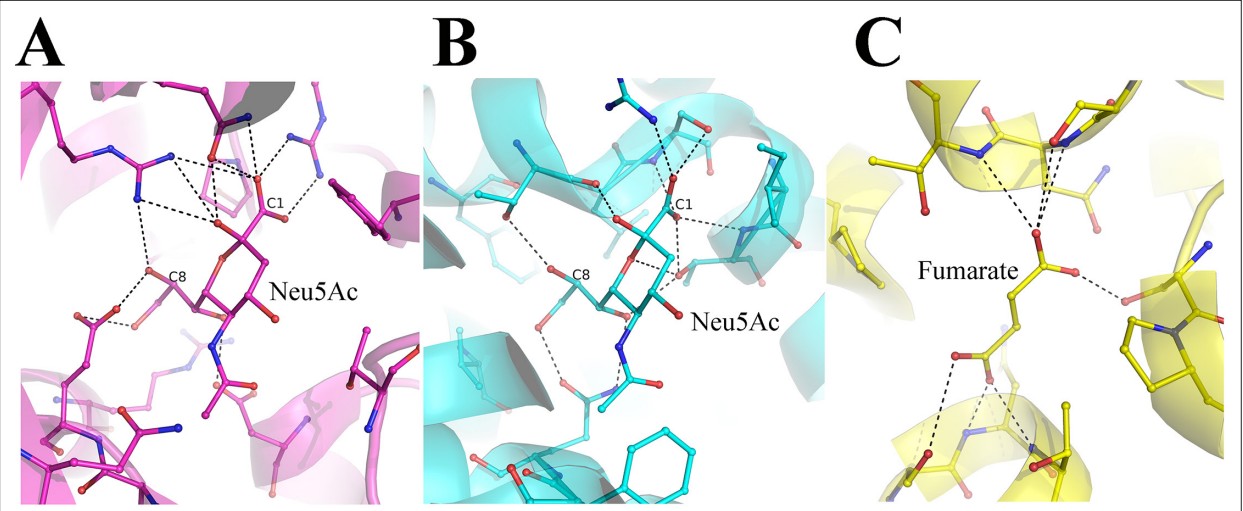

**Figure 6.** Comparison of N-Acetylneuraminic acid Neu5Ac binding pocket. (**A**) The interactions of Neu5Ac with the substrate binding protein SiaP from *F.nucleatum* (PDB-ID 4MNP). The C1 and the C8 carbon atoms are labeled to show the different ends of the nine-carbon sugar. (**B**) The interaction of Neu5Ac with the SSS-type Neu5Ac transporter (PDB-ID 5NV9). (**C**) The interaction of fumarate with the dicarboxylate transporter *Vc*INDY (PDB-ID 6OKZ).

unknown, which will reveal how Neu5Ac is transferred from SiaP to SiaQM. While mechanistic questions require further investigation, the precise definition of the binding pocket provided in this work is a starting point for structure-based drug discovery.

## Materials and methods
### Construct design
The gene sequence encoding the TRAP transporter from *F. nucleatum* (NCBI Reference Sequence: WP_005902322.1) was synthesized and cloned into a pBAD vector by GeneArt. A Strep-tag was inserted in addition to the 6X-His Tag at the N-terminus to enhance the protein purity. It was also cloned into pWarf (+) vector for fluorescence-detection size-exclusion chromatography (*Hsieh et al., 2010*).

### Protein purification of FnSiaQM
Expression trials and detergent scouting were performed as described by *Hsieh et al., 2010*. For large-scale production, the *Fn*SiaQM construct in pBAD was introduced into *E. coli* strain TOP10 cells. A single colony from the transformed plate was inoculated into 150 mL of terrific broth (TB) containing 100 µg/mL ampicillin. The culture was then incubated overnight at 30 °C in a shaker incubator. The following day, 10 mL of the primary culture was transferred to a culture flask containing 1 L Terrific broth supplemented with ampicillin and incubated at 30 °C in an orbital shaker. The cultures were induced with 0.02% arabinose at an optical density (OD) of 1.5 at 600 nm and allowed to grow for an additional 18 h. Cells were harvested by centrifugation at 4000 x g for 20 min at 4 °C. The resulting cell pellet was rapidly frozen in liquid nitrogen and stored at –80 °C for future use. The cell pellet was resuspended in lysis buffer (100 mM Tris, pH 8.0, 150 mM NaCl, and 1 mM dithiothreitol (DTT)) supplemented with lysozyme, DNase, and cOmplete EDTA-free inhibitor cocktail (Roche). Lysis was achieved by two passes through a Constant cell disruptor at 20 and 28 kPsi, respectively. Subsequently, cell debris and unbroken cells were pelleted by centrifugation at 10,000 x g for 30 min at 4 °C. The supernatant from the low-speed centrifugation was subjected to ultracentrifugation at 150,000 x g at 4 °C for 1 hr to pellet the cell membranes. The membranes were then resuspended in solubilizing buffer (1% n-dodecyl-β-D-maltoside (DDM), 50 mM Tris pH 8.0, 150 mM NaCl, 10 mM imidazole, 3 mM β-ME) at a 1:10 w/v ratio and solubilized at 4 °C for 2 hr. The insoluble membrane components were separated by ultracentrifugation for 30 min at 100,000×g, and the resulting supernatant was loaded onto a 5 mL His-Trap (Cytiva) column pre-equilibrated with buffer A (50 mM Tris

pH 8.0, 150 mM NaCl, 10 mM imidazole, 3 mM β-ME, 0.02% DDM). The resin was extensively washed with buffer A until a stable UV absorbance was achieved in the chromatogram. The *Fn*SiaQM protein was eluted using an elution buffer (50 mM Tris, pH 8.0, 150 mM NaCl, 150 mM imidazole, 3 mM β-ME, and 0.02% DDM). The eluted fractions were pooled and used for strep-tag purification. A 5 mL Strep-Trap HP column (Cytiva) was used for subsequent purification. The column was pre-equilibrated with a buffer (100 mM Tris, pH 8.0, 150 mM NaCl, 10 mM imidazole, 3 mM β-ME, and 0.02% DDM). After loading the sample onto the column, the column was washed with 10 CV buffer (100 mM Tris pH 8.0, 150 mM NaCl, 10 mM imidazole, 3 mM β-ME, and 0.02% DDM). The *Fn*SiaQM protein was eluted in the presence of 10 mM desthiobiotin, 100 mM Tris (pH 8.0), 150 mM NaCl, 3 mM β-ME, and 0.02% DDM. The eluted fraction was concentrated using a 100 kDa Amicon filter and subsequently injected into a size-exclusion column (Superdex200 16/300 increase) in buffer (50 mM Tris pH 8.0, 150 mM NaCl, 1 mM DTT, 0.02% DDM). Size exclusion chromatography yielded a monodisperse peak for *Fn*SiaQM protein (*Figure 1—figure supplement 1*).

## Generation, isolation, and purification of FnSiaQM-specific nanobodies

Generation of the nanobodies was performed by the University of Kentucky Protein Core, following previously established protocols (*Chow et al., 2019*). Alpacas were subcutaneously injected with 100 μg of recombinant *Fn*SiaQM in DDM once a week for 6 wk. Peripheral blood lymphocytes were isolated from alpaca blood and used to construct a bacteriophage display cDNA library. Two rounds of phage display against the recombinant *Fn*SiaQM identified two potentially VHH-positive clones. Positive clones were confirmed by sequencing and were analyzed for nanobody components. These two nanobodies were cloned into the pMES4 vector and *E. coli* strain BL21 (DE3) was used for their expression. A single colony was inoculated into 100 mL LB medium and cultured overnight. These overnight cultures were diluted in 1 L of LB media in a 1:100 ratio. Expression was induced with 0.5 mM Isopropyl β-D-1-thiogalactopyranoside (IPTG) when the O.D.$_{600}$ reached 1.0, followed by further incubation at 28 °C for 16 hr for protein production. Cultures were then centrifuged at 4000 × g for 20 min at 4 °C, and periplasmic extracts were prepared using the osmotic shock method with 20% sucrose. The periplasmic extract was dialyzed to remove sucrose and filtered for affinity chromatography. This filtered fraction was loaded onto a 5 mL His-Trap (Cytiva) column pre-equilibrated with buffer A (50 mM Tris pH 8.0, 150 mM NaCl, 10 mM imidazole, and 3 mM β-ME). The column was extensively washed with 20 CV of buffer B until stable UV absorbance was observed in the chromatogram. The nanobodies were eluted using an elution buffer (50 mM Tris, pH 8.0, 150 mM NaCl, 150 mM imidazole, and 3 mM β-ME). The eluted fractions were pooled and concentrated using a 10 kDa Amicon (Sigma) filter and subsequently injected into a Superdex75 column. Both nanobodies exhibited monodisperse peaks during the size-exclusion chromatography. The peak fractions were pooled and stored at 4 °C.

## Complex formation of FnSiaQM-nanobody (T4)-nanodisc (MSP1D1)

MSP1D1 was expressed and purified following established protocols (*Denisov and Sligar, 2016*). The 6X-His tag was cleaved from the affinity-purified MSP1D1 using TEV protease and the resulting product was concentrated to 5 mg/mL. For nanodisc formation, nickel affinity-eluted *Fn*SiaQM, MSP1D1, and *E. coli* polar lipids were combined in a molar ratio of 1:6:180 and incubated on ice for 15 min. Nanodisc formation was initiated by adding washed Bio-Beads (200 mg dry weight per 1 mL of the protein mixture; Bio-Rad), and the mixture was rotated for 2 hr at 4 °C. The protein mixture was subsequently separated from the Bio-Beads using a fine needle. A two-molar excess of the T4-Nanobody was added at this step to the *Fn*SiaQM-nanodiscs. A Strep-Tag chromatography step removed the empty nanodiscs and excess T4-nanobodies. The *Fn*SiaQM-reconstituted nanodisc with the T4 nanobody were then separated from the aggregates using a Superdex200 16/300 increase size exclusion column. The fractions corresponding to the peaks were concentrated to 2.5 mg/mL and used for cryo-EM analysis. To acquire the Neu5Ac-bound structure, 30 mM Neu5Ac was incorporated in the gel filtration buffer (100 mM Tris, pH 8.0, 150 mM NaCl, and 1 mM DTT).

## Cryo-EM sample preparation and data collection

The *Fn*SiaQM-T4 complex at a concentration of 2.5 mg/mL was immediately applied onto UltrAuFoil R0.6/1.0 grids (300 mesh) previously subjected to glow discharge for 150 s at 25 mA. Excess fluid

was removed by blotting using a Vitrobot Mark IV (Thermo Fisher Scientific) apparatus, and the grids were rapidly vitrified by plunging them into liquid ethane that was pre-cooled with liquid nitrogen. Subsequently, the grids were screened at a magnification of 59,000x using a Titan Krios microscope G3i (ThermoFishser Scientific) operating at 300 kV. Data were collected at a magnification of 75,000x, corresponding to a pixel size of 1.07 Å using a Falcon 3 detector in counting mode.

## Data processing of unliganded form

A total of 960 movies were collected for the unliganded structure. All processing was performed using the CryoSPARC software suite (*Punjani et al., 2017*). After patch motion correction and CTF correction, the summed images were examined and 941 micrographs were selected for particle picking. The initial particle picking (blob picker) was performed using 315 images. This was followed by 2D classification, the selection of 2D classes, and *ab initio* reconstruction into two classes. One class clearly showed the presence of the nanodisc, the protein inside, and the bound nanobody. This was refined with 49,659 particles to a resolution of 6.7 Å resolution. This 3-D map was used to create templates, template-based particle picking was performed on all 941 images, and 385,668 particles were picked. Multiple rounds of 2-D classification and selection yielded 141,272 good particles. These particles were used for *ab initio* reconstruction in cryoSPARC (*Punjani et al., 2017*), followed by homogenous and non-uniform refinement (*Punjani, 2020*). The final map resulted in an overall resolution of 3.2 Å resolution (Fourier Shell Correlation cutoff 0.143). The B-factor estimated from the Guinier plot is 125.2 Å$^2$.

## Data processing of Neu5Ac-bound form

A total of 2341 images were collected. After manually inspecting the micrographs, 1752 images were selected for further processing. The template generated from the unliganded maps was used for the particle picking. After multiple rounds of 2D classification and particle pruning, 653,554 particles were used to create six *ab initio* classes. Four of these classes appeared identical (225,006 particles) and were refined to better than 4.5 Å using homogeneous refinement. The final map was constructed using non-uniform refinement protocols in Cryo-SPARC. The map has an overall resolution of 3.17 Å at the FSC 0.143 threshold. The B-factor estimated from the Guinier plot is 135.5 Å$^2$.

## Model building and model refinement

The starting model was constructed using Alphafold2 software (*Evans et al., 2022*). Subsequently, this model was docked within the cryo-EM map of *Fn*SiaQM unliganded form of *Fn*SiaQM. A series of iterative rounds of model building using Coot (*Emsley and Cowtan, 2004*) and refinement using Phenix (*Adams et al., 2010*) were performed to complete the model. These steps were essential for enhancing the accuracy and quality of the structural model and ensuring its congruence with the experimental cryo-EM data. The refined unliganded model was used as the starting model for the substrate-bound map, and sugars and ions were modeled. All the figures were either made in UCSF Chimera or PyMOL (*DeLano, 2002*; *Pettersen et al., 2004*). The final details of the data collection, processing, and results of model building and refinement are summarized in *Table 1*.

## Reconstitution of FnSiaQM into proteoliposomes

Purified *Fn*SiaQM was reconstituted using a detergent removal method performed in batches following previously described procedures (*Wahlgren et al., 2018*). In summary, 50 µg *Fn*SiaQM was combined with 120 µL of 10% $C_{12}E_8$ detergent and 100 µL of 10% egg yolk phospholipids (w/v) in the form of sonicated liposomes. To this mixture, 50 mM K$^+$-gluconate and 20 mM HEPES/Tris (pH 7.0) were added to a final volume of 700 µL. The reconstitution blend was then exposed to 0.5 g of Amberlite XAD-4 resin while continuously stirred at 1200 rev/min at 23 °C for 40 min.

## Transport measurements and transport assay

Following reconstitution, 600 µL of proteoliposomes were loaded onto a Sephadex G-75 column (0.7 cm diameter ×15 cm height) pre-equilibrated with 20 mM HEPES/Tris (pH 7.0) containing 100 mM sucrose to balance the internal osmolarity. Valinomycin (0.75 µg/mg phospholipid) prepared in ethanol was introduced into the eluted proteoliposomes to create a K$^+$ diffusion potential. Following a 10 s incubation with valinomycin, transport was initiated by adding 5 µM [$^3$H]-Neu5Ac, 0.5 µM

*Fn*SiaP, and 10 mM Na$^+$-gluconate to 100 µL of liposomes. The initial transport rate was determined by halting the reaction after 15 min, which fell within the initial linear range of [$^3$H]-Neu5Ac uptake into the proteoliposomes, as established through the time-course experiments.

The transport assay was concluded by loading each proteoliposome sample (100 µL) onto a Sephadex G-75 column (0.6 cm diameter ×8 cm height) to eliminate external radioactivity. The experimental values were corrected by subtracting the control, i.e., the radioactivity taken up in liposomes reconstituted in the absence of protein. The radioactivity associated with the control samples, i.e., empty liposomes was less than 10% with respect to proteoliposomes. Proteoliposomes were eluted using 1 mL of 50 mM NaCl and the collected eluate was mixed with 4 mL of scintillation mixture, followed by vortexing and counting. Data analysis was conducted using Grafit software (version 5.0.13) using the first-order equation for time-course analysis. As specified in the figure legend, all measurements are presented as mean±s.d. from at least three independent experiments.

## Acknowledgements

We would like to acknowledge the University of Kentucky's Nanobody Production Facility for its assistance. We also thank Prof. Jeff Abramson (UCLA) for providing us with the pWARF vector.PG, KRV, and SR acknowledge the Department of Biotechnology for the cryo-EM and computing facility, funded under the B-Life grant DBT/PR12422/MED/31/287/2014. SR and RF, acknowledge funding from the Indo-Swedish collaborative grant from the DBT and the Swedish Research Council (BT/IN/Sweden/41/2013). This work was also supported by the DBT/Wellcome Trust India Alliance Fellowship (grant number IA/E/16/1/502999) awarded to PG. RCJD acknowledges the following for funding support: (1) the Marsden Fund Council from Government funding, managed by Royal Society Te Apārangi (contracts UOC1506 and UOC2211); (2) a Ministry of Business, Innovation, and Employment Smart Ideas grant (contract UOCX1706); and (3) the Biomolecular Interactions Center (UC).

## Additional information

### Funding

| Funder | Grant reference number | Author |
|---|---|---|
| Wellcome Trust/DBT India Alliance | IA/E/16/1/502999 | Parveen Goyal |
| Department of Biotechnology, Ministry of Science and Technology, India | DBT/PR12422/MED/31/287/2014 | Subramanian Ramaswamy |
| Department of Biotechnology, Ministry of Science and Technology, India | BT/IN/Sweden/41/2013 | Subramanian Ramaswamy |
| Marsden Fund | UOC1506 | Renwick CJ Dobson |
| Marsden Fund | UOC2211 | Renwick CJ Dobson |
| Ministry of Business, Innovation and Employment | UOCX1706 | Renwick CJ Dobson |

The funders had no role in study design, data collection and interpretation, or the decision to submit the work for publication. For the purpose of Open Access, the authors have applied a CC BY public copyright license to any Author Accepted Manuscript version arising from this submission.

### Author contributions

Parveen Goyal, Validation, Investigation, Methodology, Writing – original draft, Writing – review and editing, Funding acquisition, Project administration, Data curation; KanagaVijayan Dhanabalan, Formal analysis, Investigation, Methodology, Writing – original draft, Writing – review and editing;

Mariafrancesca Scalise, Formal analysis, Investigation; Rosmarie Friemann, Renwick CJ Dobson, Conceptualization, Formal analysis, Supervision, Writing – review and editing; Cesare Indiveri, Conceptualization, Data curation, Supervision, Investigation, Methodology, Writing – review and editing; Kutti R Vinothkumar, Conceptualization, Supervision, Investigation, Writing – review and editing; Subramanian Ramaswamy, Conceptualization, Formal analysis, Supervision, Funding acquisition, Investigation, Methodology, Writing – original draft, Writing – review and editing

### Author ORCIDs
Parveen Goyal https://orcid.org/0000-0002-7808-8298
KanagaVijayan Dhanabalan https://orcid.org/0000-0001-8636-2616
Renwick CJ Dobson https://orcid.org/0000-0002-5506-4939
Kutti R Vinothkumar https://orcid.org/0000-0002-6746-5684
Subramanian Ramaswamy https://orcid.org/0000-0002-6709-190X

Reviewer #3 (Public review): https://doi.org/10.7554/eLife.98158.4.sa1
Author response https://doi.org/10.7554/eLife.98158.4.sa2

## Additional files

### Supplementary files
MDAR checklist

### Data availability
The cryo-EM maps and the coordinates of the Neu5Ac bound and the unliganded transporter have been deposited in the Electron Microscopy Data Bank (ID 38926 and 38925), and in the Protein Data Bank (8Y4X and 8Y4W), respectively.

The following datasets were generated:

| Author(s) | Year | Dataset title | Dataset URL | Database and Identifier |
|---|---|---|---|---|
| Goyal P, Ramaswamy S, Vinothkumar KR | 2024 | Apo form of Tripartite ATP-independent Periplasmic (TRAP) transporter from Fusobacterium nucleatum | https://www.ebi.ac.uk/emdb/EMD-38926 | Electron Microscopy Data Bank, 38926 |
| Goyal P, Ramaswamy S, Vinothkumar KR | 2024 | Sialic acid bound form of Tripartite ATP-independent Periplasmic (TRAP) transporter from Fusobacterium nucleatum | https://www.ebi.ac.uk/emdb/EMD-38925 | Electron Microscopy Data Bank, 38925 |
| Goyal P, Ramaswamy S, Vinothkumar KR | 2024 | Apo form of Tripartite ATP-independent Periplasmic (TRAP) transporter from Fusobacterium nucleatum | https://www.rcsb.org/structure/8Y4X | RCSB Protein Data Bank, 8Y4X |
| Goyal P, Ramaswamy S, Vinothkumar KR | 2024 | Sialic acid bound form of Tripartite ATP-independent Periplasmic (TRAP) transporter from Fusobacterium nucleatum | https://www.rcsb.org/structure/8Y4W | RCSB Protein Data Bank, 8Y4W |

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
