## [Editor Report · eLife Assessment]

This **valuable** work provides novel insights into the substrate binding mechanism of a tripartite ATP-independent periplasmic (TRAP) transporter, which may be helpful for the development of specific inhibitors. The structural analysis is **convincing**, but additional work will be required to establish the transport mechanism as well as well as binding sites for all ligands. This study will be of interest to the membrane transport and bacterial biochemistry communities.

---

## [Referee Report · Reviewer #3 (Public review)]

The manuscript by Goyal et al report substrate-bound and substrate-free structures of a tripartite ATP independent periplasmic (TRAP) transporter from a previously uncharacterized homolog, F. nucleatum. This is one of most mechanistically fascinating transporter families, by means of its QM domain (the domain reported in his manuscript) operating as a monomeric 'elevator', and its P domain functioning as a substrate-binding 'operator' that is required to deliver the substrate to the QM domain; together, this is termed an 'elevator with an operator' mechanism. Remarkably, previous structures had not demonstrated the substrate Neu5Ac bound. In addition, they confirm the previously reported Na+ binding sites, and report a new metal binding site in the transporter, which seems to be mechanistically relevant. Finally, they mutate the substrate binding site and use proteoliposomal uptake assays to show the mechanistic relevance of the proposed substrate binding residues.

Strengths:

The structures are of good quality, the presentation of the structural data has improved, the functional data is robust, the text is well-written, and the authors are appropriately careful with their interpretations. Determination of a substrate bound structure is an important achievement and fills an important gap in the 'elevator with an operator' mechanism.

Weaknesses:

Although the possibility of the third metal site is compelling, I do not feel it is appropriate to model in a publicly deposited PDB structure without directly confirming experimentally. The authors do not extensively test the binding sites due to technical limitations of producing relevant mutants; however, their model is consistent with genetic assays of previously characterized orthologs, which will be of benefit to the field.

---

## [Author Response]

The following is the authors’ response to the previous reviews.

**Reviewer #1 (Public review):**
Summary:This manuscript reports the substrate-bound structure of SiaQM from F. nucleatum, which is the membrane component of a Neu5Ac-specific Tripartite ATP-dependent Periplasmic (TRAP) transporter. Until recently, there was no experimentally derived structural information regarding the membrane components of TRAP transporter, limiting our understanding of the transport mechanism. Since 2022, there have been 3 different studies reporting the structures of the membrane components of Neu5Ac-specific TRAP transporters. While it was possible to narrow down the binding site location by comparing the structures to proteins of the same fold, a structure with substrate bound has been missing. In this work, the authors report the Na+-bound state and the Na+ plus Neu5Ac state of FnSiaQM, revealing information regarding substrate coordination. In previous studies, 2 Na+ ion sites were identified. Here, the authors also tentatively assign a 3rd Na+ site. The authors reconstitute the transporter to assess the effects of mutating the binding site residues they identified in their structures. Of the 2 positions tested, only one of them appears to be critical to substrate binding.Strengths:The main strength of this work is the capture of the substrate bound state of SiaQM, which provides insight into an important part of the transport cycle.Weaknesses:The main weakness is the lack of experimental validation of the structural findings. The authors identified the Neu5Ac binding site, but only test 2 residues for their involvement in substrate interactions, which is quite limited. However, comparison with previous mutagenesis studies on homologues supports the location of the Neu5Ac binding site. The authors tentatively identified a 3rd Na+ binding site, which if true would be an impactful finding, but this site was not sufficiently experimentally tested for its contribution to Na+ dependent transport. This lack of experimental validation prevents the authors from unequivocally assigning this site as a Na+ binding site. However, the reporting of these new data is important as it will facilitate follow up studies by the authors or other researchers.Comments on revisions:Overall, the authors have done a good job of addressing the reviewers' comments. It's good to know that the authors are working on the characterisation of the potential metal binding site mutants - characterizing just a few of these will provide much-needed experimental support for this potential Na+ site.The new MD simulations provide additional support for the new Na+ site and could be included.However, as the authors know, direct experimental characterisation of mutants is the ideal evidence of the Na+ site.Aside from the characterisation of mutants, which seems to be held up by technical issues, the only remaining issue is the comparison of the Na+- and Na+/Neu5Ac-bound states with ASCT2. It still does not make sense to me why the authors are not directly comparing their Na+ only and Na+/Neu5Ac states with the structures of VcINDY in the Na+-only and Na+/succinate bound states. These VcINDY structures also revealed no conformational changes in the HP loops upon binding succinate, as the authors see for SiaQM. Therefore, this comparison is very supportive. It is understood that the similarity to the DASS structure is mentioned on p.17, but it is also interesting and useful to note that TRAP and DASS transporters also share a lack of substrateinduced local conformational changes, to the extent these things have been measured.

We acknowledge the summary weakness that experimental data to support the third Na binding site is critical.

Based on the reviewer’s suggestion, we added the following in the main text and a supplementary figure comparing the Na ion binding sites between VcINDY and SiaQM. Page 13.

“These two sodium ion binding sites are also conserved in the structure of VcINDY (Supplementary Figure 7) (Sauer et al., 2022). In both cases, the sodium ions are bound at the helix-loop-helix ends of HP1 and HP2. The binding sites utilize both side chains and main chain carbonyl groups. The number of main chain carbonyl interactions suggests that they are critical, and using main chain rather than side chain interactions minimizes the likelihood of point mutations affecting the binding.”

**Reviewer #3 (Public review):**
The manuscript by Goyal et al report substrate-bound and substrate-free structures of a tripartite ATP independent periplasmic (TRAP) transporter from a previously uncharacterized homolog, F. nucleatum. This is one of most mechanistically fascinating transporter families, by means of its QM domain (the domain reported in his manuscript) operating as a monomeric 'elevator', and its P domain functioning as a substrate-binding 'operator' that is required to deliver the substrate to the QM domain; together, this is termed an 'elevator with an operator' mechanism.Remarkably, previous structures had not demonstrated the substrate Neu5Ac bound. In addition, they confirm the previously reported Na+ binding sites, and report a new metal binding site in the transporter, which seems to be mechanistically relevant. Finally, they mutate the substrate binding site and use proteoliposomal uptake assays to show the mechanistic relevance of the proposed substrate binding residues.Strengths:The structures are of good quality, the presentation of the structural data has improved, the functional data is robust, the text is well-written, and the authors are appropriately careful with their interpretations. Determination of a substrate bound structure is an important achievement and fills an important gap in the 'elevator with an operator' mechanism.Weaknesses:Although the possibility of the third metal site is compelling, I do not feel it is appropriate to model in a publicly deposited PDB structure without directly confirming experimentally. The authors do not extensively test the binding sites due to technical limitations of producing relevant mutants; however, their model is consistent with genetic assays of previously characterized orthologs, which will be of benefit to the field. Finally, some clarifications of EM processing would be useful to readers, and it would be nice to have a figure visualizing the unmodeled lipid densities - this would be important to contextualize to their proposed mechanism.
**Reviewer #3 (Recommendations for the authors):**
I appreciate the authors' responses to our critiques; the revised manuscript is much improved and has addressed most of my concerns. I look forward to seeing their follow up experiments testing mutational e=ects. I think MD simulations of ion-binding sites on their own are supportive but by themselves not su=icient to prove the existence of a functional Na+-binding site. Some clarifications in the methods/supplements would satisfy my concerns about data processing and analysis.- Unliganded map: were the 141,272 particles used for one class of ab initio? This is unusual, usually multiple ab initio classes are used to further eliminate junk particles. The authors themselves use 6 classes for the substrate-bound dataset.

We classified the particles into multiple 3-D classes. There was no improvement in statistics or maps on splitting these further. Hence, we did not pursue that further.

- Substrate-bound map: how did the four 'identical' classes independently refine? Are similar Na+/substate densities found in each separate class?

The other classes refined to worse than 4.5 Å resolution. We stopped characterizing them past that point. We were hoping to see multiple conformations that are diLerent – and hopefully a class where only two sodium ions could be bound. However, any interpretation at 4.5 Å would be unreliable.

- Both maps: all ab initio classes prior to final refinement should be displayed in the supplementary workflow, this is common for EM processing diagrams.

We agree it is common – however, unless there is a good reason to discuss the other classes, we are not convinced of the value of crowding the figures.

- What specific refinement package and version of Phenix are the authors using? It seems unusual that it is not possible to refine without a metal in Phenix real-space refinement, I have seen many structures where there is no issue refining without critical ions/waters. The authors should double check that they are using the appropriate scattering table for cryo-EM, which should be "electron".

Sorry for the confusion – we did not mean to say we cannot refine without a metal. If we want to add something to the density, we cannot refine it without suggesting a metal or solvent. The site without anything added will refine without any issues but in the absence of additional verification, we cannot be sure of the identity of the ions. We are confident of the metal binding site – but not confident of the exact metal bound. We used Sodium as our first hypothesis.

We don’t think the scattering factors will help in the identification of the ions. Servalcat as part of CCP-EM can produce diLerence maps and we believe that for identification of ions, it will require higher resolution (<2.5 Å) but at this resolution, we can say that there is a nonprotein density but not more than that. We were using “electron” (which we believe is default with phenix.real_space_refine). The refinement was performed using standard protocols and appropriate scattering factors (Phenix version 1.19x), and we have previously used similar refinement protocols for other maps/models (Example -Vinothkumar KR, Arya CK, Ramanathan G, Subramanian R. 2021. Comparison of CryoEM and X-ray structures of dimethylformamidase. *Progress in Biophysics and Molecular Biology*, CryoEM microscopy developments and their biological applications 160:66–78. doi:10.1016/j.pbiomolbio.2020.06.008).

To convince the reviewer of the quality of the maps, we have added figures that show the model-to-map fit of all of the main secondary structural elements in both the unliganded and the Neu5Ac bound forms.

- I certainly understand the authors' reluctance to not model the entirety of protein densities; however, I think it would be useful to highlight these densities in the global context of the protein. A common way to show this is to show the density proximal to protein chains in one color, and the remaining densities in a contrasting color (Figure 1 somewhat demonstrates this but it is di=icult to tell). I think this would be a nice figure to show the presence and location of unmodeled densities.

We have modified supplementary figure 3 to include unmodelled densities in panels G and H for both structures.

- Small detail, "uniform" is misspelled as "unifrom" in supplementary Figure 3.

Thank you. Corrected.